# Improving the university teaching-learning process with ECO methodology: Teachers' perceptions

**Juan-Jesús Torres-Gordillo**[1]*, **Noelia Melero-Aguilar**[2], **Jesús García-Jiménez**[1]

**1** Department of Research Methods and Diagnosis in Education, Faculty of Educational Sciences, University of Seville, Seville, Spain, **2** Department of Theory and History of Education and Social Pedagogy, Faculty of Educational Sciences, University of Seville, Seville, Spain

* juanj@us.es

**Data Availability Statement:** All relevant data are within the manuscript and its Supporting Information files.

## Abstract

This study presents the results of research focused on university teachers' perceptions of the implementation of ECO (Explore, Create, and Offer) methodology. Through teachers´ responses, the objective was to learn about the impact ECO has on both teaching and learning. The sample consists of 22 teachers from four academic fields; they implemented ECO methodology during the 2018–19 academic year with 1,350 undergraduate students and 175 Master's-level students. The participating teachers belong to five universities: Universidad de Sevilla (Spain), Universitat de Barcelona (Spain), Universidade de Vigo (Spain), Universidad de Las Palmas de Gran Canaria (Spain) and Universidad Nacional de La Plata (Argentina). An exploratory and descriptive study was carried out, and the data were gathered from an online survey filled in by the teachers. Twenty-eight cases were obtained, one for each course that was involved in the project. The mean values were analysed by running a Kruskal-Wallis H test and $E_R^2$ for the effect size. In addition, the thematic analysis method was used to analyse the teachers' perceptions while representing their opinions faithfully. The results showed that ECO methodology has a very positive effect on the personal development of the teachers. ECO is a methodology that comes to have revolutionary effects, improving the relationship between teachers and students, who strengthen their commitment to their own learning. It is also an excellent means for connecting students with the social and professional world outside of academia.

## Introduction

Higher education has changed in recent decades. Since the implementation of the Bologna Process and the European Higher Education Area (EHEA), universities have become more focused on teaching by competencies and less on the transmission of scientific knowledge as stand-alone content [1]. Education has now gone beyond the role of teaching basic instrumental skills, such as reading and writing, to also include the acquisition of practical and social skills, and empower people throughout their life cycle [2–4].

**Funding:** This innovation project number 22143 has been funded in the competitive public call of the academic year 2018-19 by the 3rd Teaching Plan of the University of Seville, action 1.2.3 Support for coordination and teaching innovation to JJTG. It has also been funded by the 6th Research Plan of the University of Seville.

This transformation has been influenced by the social needs and demands of today's labour market. In order to enter into professional life, the development of social skills, creativity, critical thinking and entrepreneurship is required [5–8]. Thus, competency-based learning has been incorporated into the higher education curriculum. The result is a more holistic vision of education, facilitating a more well-rounded form of learning that is less focused on memorising information [9].

The teaching staff plays an important role in realising this vision [10]. However, teachers sometimes show certain limitations in incorporating this model into the reality of the classroom [11]. Firstly, there are barriers in the educational institution itself [3]. Additionally, they have to cope with a research and bureaucratic workload that reduces the time they have to provide quality teaching [12]. Finally, resources and training opportunities are scarce [11,13].

However, despite these limitations, teachers are interested in improving their teaching, putting into practice competency-based learning, implementing active methodologies and working in cooperative groups with other teachers [14]. Students positively value the use of non-traditional teaching methodologies that, although they require greater student involvement, have an impact on their learning, motivation, interest and education [15,16].

These student-centred inductive methodologies share common characteristics such as active learning, teamwork and social immersion. They have arisen in response to the lack of training in competencies identified by students [7], due to the benefits of going beyond the university context [17,18] and because of the interdisciplinarity of teaching [19]. Students value activities that allow them to open up debates and apply theoretical content to practical activities [20].

Some examples of student-based methodologies are service-learning, place-based education, problem-based learning and project-based learning. The use of these inductive methodologies fosters student interest and motivation, improves critical and reflective capacity, develops social skills, and empowers ethnic minorities [3,21–23]. They also have a positive effect on the assimilation of theoretical content [24].

Additionally, such methodologies transform teachers' conceptions of education. Teachers take a more holistic view of education, taking into account the social context and the individual characteristics of each student. At the same time, using these methods moves educators away from the conviction that expert knowledge should be provided to students in a theoretical and non-personalised way. As a result, the course programme teachers employ is more flexible, is open to the different demands that may arise, and views knowledge as being in constant development [25].

One such methodology is Design Thinking (DT), which involves challenge-based learning. It differs from other methodologies in that it focuses on finding solutions, not problems [26]. It makes use of tools from design to generate alternative responses that can solve a problem, with great importance being placed on prototyping, confrontation and feedback, without neglecting emotional involvement [27]. This emotional involvement leads to students 'falling in love' with the problem and using errors as a source of inspiration for learning [28]. Among the benefits of this methodology is that it places human beings–the users—at its centre [29], which helps students become aware of social complexity and grounds them in reality in order to create solutions [30–32].

In our context, DT methodology has been adapted from five phases to three, using the ECO nomenclature (Explore, Create, Offer). The explore phase corresponds to the empathise and define stages; the create phase is analogous to the ideate and prototype phases; and, finally, the offer phase includes the implementation, testing and promotion of what has been learned and generated. During the process, students must go through all the phases, which are iterative. They start with a diagnosis of needs, using empathy and an in-depth knowledge of the target

population, then create an innovative challenge that responds to the needs identified, test it and subsequently implement it [33]. In this research, the ECO method has been implemented by 22 professors from five universities and four subject areas. The research interrogates how this process has transformed teachers' perceptions of both their own teaching and the learning achieved by students.

Although there are innovative experiences as we have mentioned, our review has not found methodological innovations that are really transforming what happens in the learning process, as ECO tries to do, and especially from a committed and conscious work of the students. In addition, another gap in the literature is that there are few studies that focus on analyzing teachers' views and perceptions and how they feel and interpret what is happening in the classroom. We want to test the ECO method and fill this gap. On the basis of these issues, the objectives of this article are as follows:

- To understand university teachers' perceptions of the impact made by the ECO method on their teaching work.

- To understand university teachers' perceptions of the impact made by the ECO method on student learning.

## Methods

### Population and sample

The population is made up of 22 university professors who used the ECO method with 1350 undergraduate and 175 Master's students during the 2018–19 academic year. The teachers are distributed between 21 modules imparted in 13 different Bachelor's-level degree courses in four subject areas, and seven modules taught in five Master's degree courses belonging to three subject areas. The participating teachers come from five different universities: Universidad de Sevilla (USev), Universitat de Barcelona (UB), Universidade de Vigo (UVigo), Universidad de Las Palmas de Gran Canaria (ULPGC) and Universidad Nacional de La Plata (UNLP) in Argentina. Sixty-eight per cent are men and 32 per cent women. Their ages range from 25 to 60. University experience was organised into three categories: 7% of participants had 0–5 years' experience, 32% were experienced teachers and 61% were senior teachers. Teacher responses to the questionnaire generated a sample of 28 cases, given that some teachers taught more than one subject. These responses were voluntary, and researchers ensured that informed consent was obtained. On top of that, this research was approved by the Ethical Committee of Experimentation in Social Sciences of the University of Seville and followed its standards.

### Instrument

The data were collected in June 2019 from an online questionnaire in which teachers shared their perceptions of their experience with the ECO method [dx.doi.org/10.17504/protocols.io.bhp5j5q6]. Each teacher was asked to freely complete one questionnaire per module, answering two fundamental questions: what benefits has he or she experienced when applying ECO as a teacher? And what benefits does he or she perceive there to be for students when applying ECO?

Cronbach's alpha coefficient for this instrument was 0,853.

### Data collection and analysis procedure

The object of this research is exploratory and descriptive [34]. Mean contrasts were performed using the Kruskal-Wallis H test, as the data are not normal. The effect size was also calculated

through the epsilon coefficient squared ($E_R{}^2$), as this is an alternative to eta squared ($eta^2$) in small populations [35,36].

This is a more conservative estimator that avoids bias, according to the formula:

$$ER^2 = \frac{H}{(n2-1)/(n+1)}$$

Additionally, the Thematic Analysis Method was used [37,38], with the aim of analysing the perceptions of university teaching staff while trying to represent the contributions made as faithfully as possible. The method can be synthesised into five steps [39]:

1. An initial exploration of the data is performed, reading through all the answers and subsequently analysing them.

2. The first round of coding is carried out, assigning labels to each block of significant answers.

3. A search for patterns is performed, selecting possible themes.

4. The patterns are reviewed, adjusting and eliminating categories.

5. The final categories are determined.

Table 1 shows the process of category analysis.

Three researchers participated in the analysis process. Two initial meetings were held to read through all of the teachers' responses and analyse data iteratively so as to categorise it. Once familiar with the inductively generated categories, each researcher autonomously coded the first three questionnaires in their entirety. From the third meeting onwards, attention was paid to the reliability of the work carried out. For this purpose, the consistency of the codes used was calculated [39,40] using the Kappa Fleiss technique [41].

In these first three meetings, adjustment of the themes and categories was carried out. A total of four meetings were held until codification was fully completed. The calculations were made using version 24 of the SPSS program, applying the syntax of Kappa Fleiss MKAPPASC.

**Table 1. Categories analysed.**

| Questions | Categories | Indicators |
|---|---|---|
| RQ1. University teachers' perceptions of the impact of the ECO method on teaching | Personal and professional development | • Development of creativity.<br>• Motivation to engage with teaching.<br>• Satisfaction with the method when the process ends and the products are evaluated. |
| | Methodological revolution | • The role of the teacher in facilitating and providing support in the implementation of participative dynamics inside and outside the classroom.<br>• Collaborative teams of teachers. |
| | Improved interaction with students | • Strengthens bonds of trust with the students.<br>• Better knowledge of students' needs and interests. |
| RQ2. University teaching staff's perceptions of the impact made by the ECO method on student learning. | Transformation of the student body | • Student awareness of acquiring skills for their personal and professional development.<br>• Development of creativity.<br>• Motivation to learn.<br>• Satisfaction with the products created. |
| | Student commitment to their learning | • Self-regulation of learning.<br>• Students are aware of the gains made by working as a team.<br>• Collaborative work, dialogue and conflict resolution. |
| | Connection with the social and professional world | • Knowledge and identification of needs in the social context.<br>• Self-visualisation as professionals. |

**Table 2. Kappa Fleiss calculation.**

|  | Kappa (K) | ASE | Z-Value | P-Value |
|---|---|---|---|---|
| Meeting 3 | .9162472 | .00617351 | 56.48912967 | .000000 |
| Meeting 4 | .9277346 | .00634376 | 60.43187436 | .000000 |

K = Kappa value; ASE = asymptotic standard error; Z = standardised values; P-value = significance 2-tailed.

SPS designed by David Nichols. Prior training of the research team to internalise the meaning of the categories enabled an 'excellent' degree of agreement, according to Fleiss [41] or 'very good' according to Altman [42], to be reached in the third and fourth meetings (see Table 2).

## Results

The results shown below respond to the two objectives set for the research: 1) The university teaching staff's perceptions of the impact of the ECO method on teaching; 2) The university teaching staff's perceptions of the impact made by the ECO method on students. First, we present the contrasts that were statistically significant when applying the Kruskal-Wallis H test, together with the effect size by including the $E_R^2$ values (see Table 3).

The statistical results indicate that there are significant moderate differences (according to the value of $E_R^2$) between teaching experience and the initial real expectations that teachers had regarding improvement of the teaching-learning process by using ECO. We can affirm, with a confidence level of 95%, that senior teachers think—to a greater extent, at least, than those with less than five years of experience—that their initial expectations were higher.

There are also moderate differences (.372 as the $E_R^2$ value) between teachers of different levels of experience in the observed impact on student learning. With a confidence level of 99%, teachers with an experience of 6–10 years say—to a greater extent, at least, than those who are beginning their university careers—that they have observed a greater impact on student learning.

**Table 3. Kruskal-Wallis H test results.**

| Indep. Variables | Items | Values | N | Mean rank | Chi-square | Sig. | $E_R^2$ |
|---|---|---|---|---|---|---|---|
| Teaching experience | Initial real expectations regarding improvement of the teaching-learning process using ECO | New (0–5) | 2 | 7.50 | 6.572 | .037 | .252 |
|  |  | Experienced (6–10) | 9 | 10.39 |  |  |  |
|  |  | Senior (>10) | 17 | 17.50 |  |  |  |
|  | Observed impact on student learning | New (0–5) | 2 | 12.25 | 9.708 | .008 | .373 |
|  |  | Experience (6–10) | 9 | 15.83 |  |  |  |
|  |  | Senior (>10) | 17 | 14.06 |  |  |  |
| Year of studies | Difficulties encountered with ECO | 1st | 8 | 8.88 | 10.554 | .032 | .405 |
|  |  | 2nd | 3 | 9.17 |  |  |  |
|  |  | 3rd | 8 | 16.31 |  |  |  |
|  |  | 4th | 3 | 19.67 |  |  |  |
|  |  | Master | 6 | 19.67 |  |  |  |
| Implementation of ECO | Observed impact on professional development as a teacher | One specific topic or less | 1 | 5.00 | 11.191 | .011 | .430 |
|  |  | A thematic block | 2 | 10.25 |  |  |  |
|  |  | A complete module | 21 | 15.57 |  |  |  |
|  |  | Other | 4 | 13.38 |  |  |  |

There are equally moderate differences, according to the effect size ($E_R^2$ = .405), with a confidence level of 95%, between the difficulties encountered with the method and the year of studies in which ECO is implemented. Teaching staff of higher levels (fourth year undergraduates and Master's students) are shown–to a greater extent, at least, than those teaching the first year–to have greater difficulties when working with ECO. This could be due to the fact that, in advanced courses, students place greater demands on teachers. They have more knowledge, which requires more work of the teacher, and they have spent more time in traditional teaching contexts, which makes it more difficult for them to work with an alternative method.

Finally, the degree to which ECO is implemented determines the level of impact on the professional development of teachers. With a confidence level of 95%, and a moderate $E_R^2$ close to .5, we can affirm that teachers who apply ECO to the whole subject assess the impact on their own training more positively, at least, than those who only use it in one module or less. This result allows us to conclude that the ECO method would be a positive training opportunity for teachers who are keen to try out innovative learning methodologies.

## University teachers' perceptions of the impact made by the ECO method on teaching work

The results show that implementation of the ECO method has a very positive impact on teachers in terms of their personal development. The participants in our study generally believe that ECO allows them to teach with greater creativity. They generate new ideas and are more open-minded towards the teaching-learning process, which enables them to improvise and be creative, drawing on the visions and concerns expressed by students when working on challenges ("It has allowed me to develop my creativity as a teacher [. . .], generating new ideas with them" [Module 16]).

However, the teachers recognise that greater commitment and dedication to teaching is required, which sometimes means a greater workload, but the quality of the challenges faced by the students and the products created by the end of the modules generate great satisfaction in the teaching staff, which compensates for the effort put in ("Reaching the end of the subject and seeing the progress of the students, seeing the teams' end results and what they have achieved, and sharing it with everybody, is very gratifying" [Module 10]). Part of this satisfaction can be seen in the increase in the number of students on the teams.

The ECO method enables teachers to be innovative in how they run their classes. It allows them to rethink how the modules are focused and transform knowledge into action, which generates greater motivation when dealing with classes ("I am motivated because I believe that students will learn in a productive and fun way" [Module 5]).

Beyond the limitations of the university learning context, with overcrowded classrooms and traditional furniture such as benches and chairs that are fixed to the floor, the ECO method represents a methodological revolution. With regard to the way in which they teach, the teachers highlight the change involved in focusing the teaching-learning process on students and giving them a more central role, in comparison with the traditional model that focused this process on the teacher.

("The greatest benefit is seeing, when you finish teaching the module, the level of commitment from teachers and students that has been generated in the process, [. . .] as a teacher, I cease to be the centre of attention and that role is taken on by the rightful person: the student" [Module 6]).

Teachers challenge the limitations of the classroom context in which they must teach. They value the use of teams and the participatory dynamics generated in class. In such classes, students learn to debate and reflect critically on their own learning, including how much they believe they have learned.

("It has allowed me to take a significant step towards a student-centred, learning-centred teaching model, as opposed to the traditional model based on the teacher and content" [Module 15]).

When the ECO method is used for the first time, this change of focus in the teaching-learning process may cause some uncertainty. Firstly, due to the lack of experience using the method and having to deal with new modules with complex content. Secondly, because in each academic year the student body is different and, sometimes, the groups are not as participative, so less likely to fit well with this type of methodology. However, teachers consider all of these issues to be strengths that lead them to develop professionally through ECO, because once the method has been implemented over various academic years, teachers cannot imagine giving classes in any other way.

("What at first I felt to be a difficulty, after so many years of doing things in a conventional way, has become a benefit, a professional strength, because now I cannot imagine giving classes in any way other than this" [Module 7]).

Implementing the ECO method also involves a change in teachers' vision of university education ("Understanding university education in another, more holistic way, with a view to improving student and teacher education, in order to meet people's real needs" [Module 12]). The participants underline how it helps them gain a more holistic vision of education, and review and reformulate how they implement their modules ("I included this module in the process of improving the USev's innovation plan: rethinking the module itself has had a positive impact on how I have approached and implemented the ECO method in it" [Module 8]).

Throughout this process, with its challenges and uncertainties, the participants point out how teamwork among the teaching staff has been very important in implementing the ECO method. Teachers had the opportunity to share their experience as they implemented the method with their students ("Having a team of teachers who are applying the same methodology, which makes you feel like you are not alone in the process" [Module 2]). This direct support in planning, developing and evaluating the method among peers from different disciplines and subject areas has been very positive and enriching.

Finally, the results obtained highlight how, through the ECO method, interaction with students is improved. The methodology makes it possible to work with students more directly and interact with them, to get to know them better and get students to commit themselves to their work ("Additionally, the good relationship with the student body, enabling bonds of trust to be established that mean that students do not want to disappoint the teacher and make them give the best of themselves" [Module 4]). By setting up teams with a small number of members, a closer relationship with the students is established, allowing teachers to respond to the needs highlighted by them more quickly and efficiently.

## University teachers' perceptions of the impact made by the ECO method on students

The results show that the ECO method leads to a transformation in the student body. This educational metamorphosis, in which students are the protagonists, makes them aware of acquiring the competencies that are so necessary for their personal and professional development. As observed of teaching staff, participants in our study shared the view that implementing this method requires students to meet the challenges of an open initiative in which creativity is key: the motor that drives the whole process ("The student took the initiative to "create" and "modulate" his learning, and saw how he progressively moved closer to achieving a final 'product' that had a personal/community validity beyond rote-learning for an examination" [Module 19]). The ECO method promotes the development of creativity in both the

design and planning of the product. In this way, it contributes to providing students with the type of well-rounded education that is underrepresented in current university curricula.

The dynamism of the method itself also generates motivation to participate and get involved, because students find meaning in what they do, can apply their creativity and engage in a varied range of activities in class ("It improves the motivation and involvement of the students, especially in Master's degree modules, where the students apply themselves" [Module 3]).

As our participants point out, in addition to developing these competencies, the ECO method generates a methodological revolution in the student learning process. While students take on a more passive role when traditional methodologies are used, the way of working when applying ECO is more autonomous and enterprising. It fosters student responsibility, decision-making and time management, while making students take a leading role in their own learning from the beginning ("It gives them the opportunity to get to know another way of learning, which is not generally the one that they are used to" [Module 18]).

Additionally, students make a commitment to their learning. As a result, they become more autonomous and responsible, expect more of themselves as well as others, and learn to listen, be more flexible in their positions and work in teams. An environment is created in which students can talk about what they think and learn to participate during class discussions ("Students make a commitment to the class and to the learning objectives, understanding that all learning takes effort" [Module 17]).

Another important aspect highlighted by the participants in the study is how the ECO method promotes students' connection with the social and professional world. It helps them to construct a critical discourse on the social world in which they find themselves, and to connect the knowledge acquired with the needs of the social context. In this way, it encourages the development of civic and social skills that are vital if students are to acquire a well-rounded, socially-committed education. For teachers, drawing students' attention to opportunities to link what they learn with the social world around them is key ("It enables them to connect with neighbourhoods and communities, to contribute the knowledge and skills acquired over the years of their degree to improving [the situation] of certain social groups" [Module 1]).

Connection with the professional world is also very central, and students see this as an opportunity to visualise themselves more as professionals than as students. The method encourages them to explore people's real needs. It helps them to see the reality of their profession in all its glory. Participants point out how, through ECO, students learn to work in teams to do a real task that is based on challenges and involves working with real people and open problems, to which they must provide the best possible solution.

## Discussion and conclusions

The results of active teaching methods are usually evaluated through the students' own testimonies, which is why studies with more in-depth techniques are necessary [43]. Therefore, the objectives of this study were to describe university teaching staff's perceptions of the impact made by the ECO method: firstly, on teaching; and secondly, on student learning. In this way, teachers' perspectives as agents involved in the teaching-learning process could be highlighted.

With regard to the first objective, we can conclude that the results show the implementation of the ECO method to have a very positive impact on teachers in terms of their personal and emotional development [25]. While it is true that some uncertainties may arise when faced with a new method, and that it is sometimes difficult to break with traditional academic routines [44], ECO becomes a strategic tool for teachers' emotional development [25]. When teachers manage to gain the necessary autonomy, they are able to undertake innovative

methodologies. In this sense, other research [45,46], agrees in pointing out how this autonomy that teachers acquire is what leads them to participate in innovative initiatives and projects that promote their personal development from the educational context. ECO boosts their creativity and their ability to engage with and generate ideas in a dynamic, original way. It motivates them to approach academic practice in an innovative way. And, above all, ECO generates great satisfaction in the teaching staff when they reach the end of the process and evaluate the scope of the challenges carried out and the products created. This conclusion is also shared by Torres-Gordillo & Herrero-Vázquez [47], who point out how teachers, in their search to improve teaching and students' professional development, satisfactorily value the innovations they make according to the impact they observe in learning and in quality improvement of their courses.

Regarding the teaching-learning process, ECO is a methodology that has revolutionary effects for teachers. The teacher gives a more central role to the students, and takes on the role of facilitator and companion, generating new dynamics in the classroom, as reflected in other studies [45,48,49]. The method promotes teamwork within and outside the university context, challenge-based learning, critical thinking and helps create a connection between theory and practice. Additionally, value is placed throughout the process on collaborative work between the teachers who are implementing the method, giving them greater security in their teaching practice [25]. This collaborative work between teachers, as asserted in similar research [50–52], allows teachers to express their doubts and reflect together on the progress made, receiving continuous feedback and feedforward, as well as sharing resources to improve teaching.

Implementation of the ECO method leads to an improvement in the relationship between teachers and students [53]. It enables the teacher to personalise how they interact with and monitor students to a greater extent. It enables them to identify the needs and specific qualities of each student, and to offer students a quicker and more effective response to the difficulties they face. It generates a climate of trust and security, helping to create environments that are more conducive to learning [20,53]. This idea is shared by Ayllón, Alsina & Colomer [54], whose research states that students obtain higher grades when their teachers show them confidence and willingness by offering them the necessary academic help and resources.

With regard to the second objective, the results lead us to conclude that the ECO method helps to transform students, providing them with a more well-rounded and socially-aware education, as other studies indicate [3,11,22]. An education in which the development of competencies is especially present. The different phases of the process drive the development of key competencies that are vital for students while being underrepresented in current university curricula [55–58]. The method's most important contribution is its ability to develop the creativity of students, as well as their interest in undertaking projects and providing original, unique and authentic solutions, as reflected in other research [59,60].

When it comes to the learning process, the ECO method requires students to make a commitment to their learning, giving them a central role in it [61]. This promotes a dynamic of teamwork on which it is based, where students are required to develop empathy, as well as engaging in dialogue and conflict resolution, as considered in other research [62,63]. In this way, through ECO, students become more autonomous and responsible for completing tasks, both individually and in groups, committing them to the process [64]. Students come to understand that effort and rigour are integral parts of engaging in the working process, providing them with great satisfaction in achieving results and final products, in which other studies coincide [65–67].

In short, the ECO method is an excellent vehicle for connecting students with the social and professional world. It helps them learn about the social context and to identify the real needs of people, groups, and social and educational entities. It encourages students to connect

with them, learn from them, and to look for real solutions and proposals, as other research indicates [68,69]. Furthermore, it allows students to visualise themselves professionally [70], connect with professionals from their field of knowledge, and become familiar with the competencies that they will exercise in their professional environment. This is highly valued by students, as reflected in the study by Villalobos-Abarca, Herrera-Acuña, Ramírez, & Cruz [71]. The ECO method helps develop civic and social commitment in students, providing them with a more socially-responsible education. This aspect is shared by other studies [22,72–74], which stress the need to go further, and urge the university to commit to the development of innovative practices with society, transferring its knowledge, strategies and values.

Finally, we conclude by pointing out the importance of promoting refreshing teaching methodologies, transformative practices and innovative projects, due to their positive impact on the quality and relevance of university education [72]. And, above all, because they manage to awaken the motivation of students, one of the main goals of any university teacher [75].

The study is limited by the fact that the data come exclusively from teaching staff, whose perceptions were expressed when completing the questionnaire presented to them. Therefore, the information analysed has not been obtained by direct experimentation. This limitation is a feature of most research in the field of education. Nevertheless, this project is being continued through a new phase in which other instruments will be used, such as interviews and discussion groups, which will allow us to broaden the conclusions obtained.

## Supporting information

**S1 Questionnaire.**
(PDF)

## Author Contributions

**Conceptualization:** Juan-Jesús Torres-Gordillo, Noelia Melero-Aguilar, Jesús García-Jiménez.

**Data curation:** Juan-Jesús Torres-Gordillo, Noelia Melero-Aguilar, Jesús García-Jiménez.

**Formal analysis:** Juan-Jesús Torres-Gordillo, Noelia Melero-Aguilar, Jesús García-Jiménez.

**Funding acquisition:** Juan-Jesús Torres-Gordillo.

**Investigation:** Juan-Jesús Torres-Gordillo, Noelia Melero-Aguilar, Jesús García-Jiménez.

**Methodology:** Juan-Jesús Torres-Gordillo, Noelia Melero-Aguilar, Jesús García-Jiménez.

**Project administration:** Juan-Jesús Torres-Gordillo, Noelia Melero-Aguilar, Jesús García-Jiménez.

**Resources:** Juan-Jesús Torres-Gordillo, Noelia Melero-Aguilar, Jesús García-Jiménez.

**Software:** Juan-Jesús Torres-Gordillo, Noelia Melero-Aguilar, Jesús García-Jiménez.

**Supervision:** Juan-Jesús Torres-Gordillo, Noelia Melero-Aguilar, Jesús García-Jiménez.

**Validation:** Juan-Jesús Torres-Gordillo, Noelia Melero-Aguilar, Jesús García-Jiménez.

**Visualization:** Juan-Jesús Torres-Gordillo, Noelia Melero-Aguilar, Jesús García-Jiménez.

**Writing – original draft:** Juan-Jesús Torres-Gordillo, Noelia Melero-Aguilar, Jesús García-Jiménez.

**Writing – review & editing:** Juan-Jesús Torres-Gordillo, Noelia Melero-Aguilar, Jesús García-Jiménez.

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
