## [Decision Letter · Decision Letter 0]

15 Jun 2020

PONE-D-19-34682

Improving the university teaching-learning process with ECO methodology: Teachers’ perceptions

PLOS ONE

Dear Dr. Torres-Gordillo,

Thank you for submitting your manuscript to PLOS ONE. After careful consideration, we feel that it has merit but does not fully meet PLOS ONE’s publication criteria as it currently stands. Therefore, we invite you to submit a revised version of the manuscript that addresses the points raised during the review process.

We look forward to receiving your revised manuscript.

Kind regards,

Haoran Xie

Academic Editor

PLOS ONE

Journal Requirements:

2. PLOS ONE will consider submissions that present new methods, software, or databases as the primary focus of the manuscript if they meet the criteria of utility, validation, and availability described here: http://journals.plos.org/plosone/s/submission-guidelines#loc-methods-software-databases-and-tools. To meet these criteria, please provide supporting materials enabling other teachers and researchers to replicate your teaching intervention such as sample worksheets, a detailed lesson plan or curriculum or other educational materials. If you include supporting materials, they should not be under a copyright more restrictive than CC-BY.”

3. Thank you for including your ethics statement:

'All participants in this research knew the nature of the study and the conditions of their participation. This participation was voluntary and followed the rules of informed

consent. This research has followed the internal regulations in the field of Social

Sciences of the Ethical Committee of Experimentation of the University of Seville

(Spain).'

Please amend your current ethics statement to confirm that your named institutional review board or ethics committee specifically approved this study.

Additional Editor Comments (if provided):

Reviewers' comments:

Reviewer's Responses to Questions

**Comments to the Author**

1. Is the manuscript technically sound, and do the data support the conclusions?

Reviewer #1: Yes

Reviewer #2: Yes

2. Has the statistical analysis been performed appropriately and rigorously? 

Reviewer #1: Yes

Reviewer #2: Yes

3. Have the authors made all data underlying the findings in their manuscript fully available?

Reviewer #1: Yes

Reviewer #2: Yes

4. Is the manuscript presented in an intelligible fashion and written in standard English?

Reviewer #1: Yes

Reviewer #2: Yes

5. Review Comments to the Author

Reviewer #1: I think this study presents a good research about the implementation of ECO (Explore, Create, and Offer) methodology. It has several bright spots. First of all, it involves a sample which consists of 22 teachers from four academic fields; and the investigators implemented ECO methodology with 1,350 undergraduate students and 175 Master’s-level students. With such a great population, it is worthwhile to be read by someone else. Apart from that, the results showed positively. Therefore, I recommend it to the editor so that ECO methodology can be spread to more educators to improve the relationship between teachers and students, who strengthen their commitment to their own learning.

Reviewer #2: This study presents the results of research focused on university teachers’ perceptions

of the implementation of ECO methodology. The question investigated in this paper is timely and interesting. The quantitative analysis appears to be technically sound, and the conclusions drawn are justified. Overall, this is a nice paper, original, largely well-written, and clearly structured, that will be a valuable contribution to the literature.

I have only very few minor suggestions for further improvement.

1. Abstract: There is no need to explain what ECO is.

2. The Introduction is not well-written. The authors uncritically reviewed previous studies without pointing out the research gap in the existing literature. Why the present study has to be conducted? How will the present study address the gap surfaced in previous studies? In other words, the rationale for the present study is not clearly stated.

3. Since questionnaire is the most important instrument for the present study, I was wondering whether the reliability of the questionnaire was checked. The authors have to state the reliability of the questionnaire. Please also include the full questionnaire in supplementary materials.

4. The Discussion section is anything but substantial. The authors did not relate the present findings with previous studies.

6. PLOS authors have the option to publish the peer review history of their article (what does this mean?). If published, this will include your full peer review and any attached files.

Reviewer #1: No

Reviewer #2: No

---

## [Author Response · Author response to Decision Letter 0]

19 Jun 2020

A. Editor's comments 

2. PLOS ONE will consider submissions that present new methods, software, or databases as the primary focus of the manuscript if they meet the criteria of utility, validation, and availability described here: http://journals.plos.org/plosone/s/submission-guidelines#loc-methods-software-databases-and-tools. To meet these criteria, please provide supporting materials enabling other teachers and researchers to replicate your teaching intervention such as sample worksheets, a detailed lesson plan or curriculum or other educational materials. If you include supporting materials, they should not be under a copyright more restrictive than CC-BY.”

3. Thank you for including your ethics statement:

'All participants in this research knew the nature of the study and the conditions of their participation. This participation was voluntary and followed the rules of informed consent. This research has followed the internal regulations in the field of Social Sciences of the Ethical Committee of Experimentation of the University of Seville (Spain).'

Please amend your current ethics statement to confirm that your named institutional review board or ethics committee specifically approved this study.

A. List of changes of each point that has been raised

1. PLOS ONE's style requirements has been thoroughly revised. 

2. The instrument used in the research has been incorporated as complementary material. This document has been translated into English. See [dx.doi.org/10.17504/protocols.io.bhp5j5q6]

3. This information has been incorporated into the text. The authors have added the following text into the main paper document:

“On top of that, this research was approved by the Ethical Committee of Experimentation in Social Sciences of the University of Seville and followed its standards”.

The authors have added this text to the “Ethics Statement” field of the submission form, by rewriting the text as follows:

“This research has followed the internal regulations in the field of Social Sciences and was approved by the Ethical Committee of Experimentation of the University of Seville (Spain).”

-------------/////---------------------/////-------------------------------------------

B. Responses to the reviewer #2

Your comments:

I have only very few minor suggestions for further improvement.

1. Abstract: There is no need to explain what ECO is.

2. The Introduction is not well-written. The authors uncritically reviewed previous studies without pointing out the research gap in the existing literature. Why the present study has to be conducted? How will the present study address the gap surfaced in previous studies? In other words, the rationale for the present study is not clearly stated.

3. Since questionnaire is the most important instrument for the present study, I was wondering whether the reliability of the questionnaire was checked. The authors have to state the reliability of the questionnaire. Please also include the full questionnaire in supplementary materials.

4. The Discussion section is anything but substantial. The authors did not relate the present findings with previous studies.

B. List of changes of each point that has been raised

1. The authors have removed this comment from the abstract: 

“ECO is an inductive methodology based on challenges and inspired by Design Thinking”, following the reviewer's recommendations.

2. The authors have reviewed the Introduction section. The modifications of the text have been highlighted in the text in grey. The authors have incorporated the following texts to the main paper document:

“Although there are innovative experiences as we have mentioned, our review has not found methodological innovations that are really transforming what happens in the learning process, as ECO tries to do, and especially from a committed and conscious work of the students. In addition, another gap in the literature is that there are few studies that focus on analyzing teachers' views and perceptions and how they feel and interpret what is happening in the classroom. We want to test the ECO method and fill this gap.”

The instrument used in the research has been incorporated as complementary material. This document has been translated into English. Also, the authors have incorporated the following text to the Instrument section: [dx.doi.org/10.17504/protocols.io.bhp5j5q6]

3. The reliability of the questionnaire was checked and incorporated into the Method section. The modifications of the text have been highlighted in the text in grey. The authors have incorporated the following text to the main paper document:

“Cronbach’s alpha coefficient for this instrument was 0,853.”

4. The authors have reviewed the Discussion section. The modifications of the text have been highlighted in the text in grey. The authors have incorporated the following texts to the main paper document:

“The results of active teaching methods are usually evaluated through the students' own testimonies, which is why studies with more in-depth techniques are necessary [43]. Therefore, the objectives of this study were to describe university teaching staff’s perceptions of the impact made by the ECO method: firstly, on teaching; and secondly, on student learning. In this way, teachers’ perspectives as agents involved in the teaching-learning process could be highlighted. “

“With regard to the first objective, we can conclude that the results show the implementation of the ECO method to have a very positive impact on teachers in terms of their personal and emotional development [25].”

“When teachers manage to gain the necessary autonomy, they are able to undertake innovative methodologies. In this sense, other research [45-46], agrees in pointing out how this autonomy that teachers acquire is what leads them to participate in innovative initiatives and projects that promote their personal development from the educational context.”

“This conclusion is also shared by Torres-Gordillo & Herrero-Vázquez [47], who point out how teachers, in their search to improve teaching and students' professional development, satisfactorily value the innovations they make according to the impact they observe in learning and in quality improvement of their courses.”

“The teacher gives a more central role to the students, and takes on the role of facilitator and companion, generating new dynamics in the classroom, as reflected in other studies [45, 48-49].”

“This collaborative work between teachers, as asserted in similar research [50-52], allows teachers to express their doubts and reflect together on the progress made, receiving continuous feedback and feedforward, as well as sharing resources to improve teaching.”

“This idea is shared by Ayllón, Alsina & Colomer [54], whose research states that students obtain higher grades when their teachers show them confidence and willingness by offering them the necessary academic help and resources.”

“With regard to the second objective, the results lead us to conclude that the ECO method helps to transform students, providing them with a more well-rounded and socially-aware education, as other studies indicate [3,11,22]. An education in which the development of competencies is especially present.”

“The method’s most important contribution is its ability to develop the creativity of students, as well as their interest in undertaking projects and providing original, unique and authentic solutions, as reflected in other research [59-60].”

“This promotes a dynamic of teamwork on which it is based, where students are required to develop empathy, as well as engaging in dialogue and conflict resolution, as considered in other research [62-63]. In this way, through ECO, students become more autonomous and responsible for completing tasks, both individually and in groups, committing them to the process [64]. Students come to understand that effort and rigour are integral parts of engaging in the working process, providing them with great satisfaction in achieving results and final products, in which other studies coincide [65-67].”

“It encourages students to connect with them, learn from them, and to look for real solutions and proposals, as other research indicates [68-69]. Furthermore, it allows students to visualise themselves professionally [70], connect with professionals from their field of knowledge, and become familiar with the competencies that they will exercise in their professional environment. This is highly valued by students, as reflected in the study by Villalobos-Abarca, Herrera-Acuña, Ramírez, & Cruz [71]. The ECO method helps develop civic and social commitment in students, providing them with a more socially-responsible education. This aspect is shared by other studies [22, 72-74], which stress the need to go further, and urge the university to commit to the development of innovative practices with society, transferring its knowledge, strategies and values.”

“Finally, we conclude by pointing out the importance of promoting refreshing teaching methodologies, transformative practices and innovative projects, due to their positive impact on the quality and relevance of university education [72]. And, above all, because they manage to awaken the motivation of students, one of the main goals of any university teacher [75].”

Also, we have added 17 new references:

43 Hartikainen S, Rintala H, Pylväs L, Nokelainen P. The concept of active learning and the measurement of learning outcomes: A review of research in engineering higher education. Educ. Sci. 2019; 9(4):276. http://dx.doi.org/10.3390/educsci9040276

45 Gavrilyuk OA, Tareva EG, Lakhno AV. Investigating the association between university teachers’ professional autonomy and their innovation performance. Pedagogika. 2019; 133(1):128–148. http://dx.doi.org/10.15823/p.2019.133.7

46 Kumpulainen K, Vierimaa SM, Koskinen-Koivisto E. Developing connective pedagogy in cultural research–A case study from the teachers’ perspective in adopting a problem-based approach in higher education. Educ. Sci. 2019; 9(4):252. http://dx.doi.org/10.3390/educsci9040252

47 Torres-Gordillo JJ, Herrero-Vázquez EA. Innovación metodológica transdisciplinar en la universidad con el método ECO. In: Reyes-Tejedor M, Cobos-Sanchiz D, López-Meneses E, Coords. Innovación pedagógica universitaria: reflexiones y estrategias. Barcelona: Octaedro; 2020. Cap. 9

50 Dawson P, Henderson M, Mahoney P, Phillips M, Ryan T, Boud D, et al. What makes for effective feedback: staff and student perspectives. Assess Eval High Educ. 2018; 44(1):25–36. https://www.doi.org/10.1080/02602938.2018.1467877

51 Kates FR, Samuels SK, Case JB, Dujowich M. Lessons Learned from a pilot study implementing a team-based messaging application (Slack) to improve communication and teamwork in Veterinary Medical Education. J Vet Med Educ. 2020; 47(1):18–26. https://www.doi.org/10.3138/jvme.0717-091r2

52 Torres-Gordillo JJ, García-Jiménez J, Herrero-Vázquez, EA. Contributions of technology to cooperative work for university innovation with Design Thinking [Aportaciones de la tecnología al trabajo cooperativo para la innovación universitaria con Design Thinking]. Píxel-Bit. 2020; 59. http://dx.doi.org/10.12795/pixelbit.74554

54 Ayllón S, Alsina Á, Colomer J. Teachers’ involvement and students’ self-efficacy: Keys to achievement in higher education. Dalby AR, Editor. PLoS ONE. 2019; 14(5). http://dx.doi.org/10.1371/journal.pone.0216865

55 Rodríguez-Martínez A, Cortés-Pascual A, Val-Blasco S. Análisis de la mejora del nivel de empleabilidad de los universitarios mediante la mejora de competencias transversales y habilidades [Analysis of the increase of the employability level in university students through the improvement of transversal]. Rev. Española Orientac. Psicopedag. 2019; 30(3):102-119. http://dx.doi.org/10.5944/reop.vol.30.num.3.2019.26275

58 Sá MJ, Serpa S. Transversal competences: their importance and learning processes by higher education students. Educ. Sci. 2018; 8(3):126. http://dx.doi.org/10.3390/educsci8030126

59 Safapour E, Kermanshachi S, Taneja P. A review of nontraditional teaching methods: Flipped classroom, gamification, case study, self-learning, and social media. Educ. Sci. 2019; 9(4):273. http://dx.doi.org/10.3390/educsci9040273

64 Jeong J, Cañada-Cañada F, González-Gómez D. The study of flipped-classroom for pre-service science teachers. Educ. Sci. 2018; 8(4):163. http://dx.doi.org/10.3390/educsci8040163

67 Valūnaitė-Oleškevičienė G, Puksas A, Gulbinskienė D, Mockienė L. Student experience on the development of transversal skills in university studies. Pedagogika. 2019; 133(1):63–77. http://dx.doi.org/10.15823/p.2019.133.4

71 Villalobos-Abarca MA, Herrera-Acuna, RA, Ramírez, IG, Cruz, XC. Aprendizaje basado en proyectos reales aplicado a la formación del ingeniero de software [Real project-based learning applied to software engineers’ education]. Form. Univ. 2018; 11(3):97-112. http://dx.doi.org/10.4067/S0718-50062018000300097

72 Pérez-Pérez C, González-González H, Lorenzo-Moledo M, Crespo-Comesaña J, Belando-Montoro MR, Costa-París A. Service-learning in Spanish universities: A study based on the perception of the dean's teams [Aprendizaje-Servicio en las universidades españolas: un estudio basado en la percepción de los equipos decanales]. RELIEVE. 2019; 25(2). http://dx.doi.org/10.7203/relieve.25.2.15029

73 Warren, S. Re-Thinking the “Problem” in Inquiry-Based Pedagogies through Exemplarity and World-Oriented. Educ. Sci. 2019; 9(4):295. http://dx.doi.org/10.3390/educsci9040295

75 Herrero-Vázquez EA, Torres-Gordillo JJ. ECO en la educación superior: un aprendizaje resonante. In: Reyes-Tejedor M, Cobos-Sanchiz D, López-Meneses E, Coords. Innovación pedagógica universitaria: reflexiones y estrategias. Barcelona: Octaedro; 2020. Cap. 8.

---

## [Decision Letter · Decision Letter 1]

3 Aug 2020

Improving the university teaching-learning process with ECO methodology: Teachers’ perceptions

PONE-D-19-34682R1

Dear Dr. Torres-Gordillo,

We’re pleased to inform you that your manuscript has been judged scientifically suitable for publication and will be formally accepted for publication once it meets all outstanding technical requirements.

Kind regards,

Haoran Xie

Academic Editor

PLOS ONE

Additional Editor Comments (optional):

Reviewers' comments:

Reviewer's Responses to Questions

**Comments to the Author**

1. If the authors have adequately addressed your comments raised in a previous round of review and you feel that this manuscript is now acceptable for publication, you may indicate that here to bypass the “Comments to the Author” section, enter your conflict of interest statement in the “Confidential to Editor” section, and submit your "Accept" recommendation.

Reviewer #1: All comments have been addressed

Reviewer #2: All comments have been addressed

2. Is the manuscript technically sound, and do the data support the conclusions?

Reviewer #1: Yes

Reviewer #2: Yes

3. Has the statistical analysis been performed appropriately and rigorously? 

Reviewer #1: Yes

Reviewer #2: Yes

4. Have the authors made all data underlying the findings in their manuscript fully available?

Reviewer #1: Yes

Reviewer #2: Yes

5. Is the manuscript presented in an intelligible fashion and written in standard English?

Reviewer #1: Yes

Reviewer #2: Yes

6. Review Comments to the Author

Reviewer #1: Thank you for revising，I checked every point of changes you made，and I think you did a nice work.

You have adequately addressed my comments raised in a previous round of review and I think that this manuscript is now acceptable for publication.

Reviewer #2: (No Response)

7. PLOS authors have the option to publish the peer review history of their article (what does this mean?). If published, this will include your full peer review and any attached files.

Reviewer #1: No

Reviewer #2: No

---

## [Editor Report · Acceptance letter]

5 Aug 2020

PONE-D-19-34682R1 

Improving the university teaching-learning process with ECO methodology: Teachers’ perceptions 

Dear Dr. Torres-Gordillo:

I'm pleased to inform you that your manuscript has been deemed suitable for publication in PLOS ONE. Congratulations! Your manuscript is now with our production department. 

Kind regards, 

on behalf of

Professor Haoran Xie 

Academic Editor

PLOS ONE